# VICRegL: Self-Supervised Learning of Local Visual Features

**Adrien Bardes**[1,2]          **Jean Ponce**[2,4]          **Yann LeCun**[1,3,4]

[1]Meta, FAIR
[2]Inria, École normale supérieure, CNRS, PSL Research University
[3]Courant Institute, New York University
[4]Center for Data Science, New York University

## Abstract

Most recent self-supervised methods for learning image representations focus on either producing a global feature with invariance properties, or producing a set of local features. The former works best for classification tasks while the latter is best for detection and segmentation tasks. This paper explores the fundamental trade-off between learning local and global features. A new method called VICRegL is proposed that learns good global and local features simultaneously, yielding excellent performance on detection and segmentation tasks while maintaining good performance on classification tasks. Concretely, two identical branches of a standard convolutional net architecture are fed two differently distorted versions of the same image. The VICReg criterion is applied to pairs of global feature vectors. Simultaneously, the VICReg criterion is applied to pairs of local feature vectors occurring before the last pooling layer. Two local feature vectors are attracted to each other if their $l^2$-distance is below a threshold or if their relative locations are consistent with a known geometric transformation between the two input images. We demonstrate strong performance on linear classification and segmentation transfer tasks. Code and pretrained models are publicly available at:
https://github.com/facebookresearch/VICRegL

## 1 Introduction

Recent advances in self-supervised learning for computer vision have been largely driven by downstream proxy tasks such as image categorization, with convolutional backbones [Chen et al., 2020a,b, Grill et al., 2020, Lee et al., 2021, Caron et al., 2020, Zbontar et al., 2021, Bardes et al., 2022, Tomasev et al., 2022], or vision transformers [Caron et al., 2021, Chen et al., 2021, Li et al., 2022, Zhou et al., 2022a]. Current approaches rely on a *joint embedding architecture* and a loss function that forces the learned features to be invariant to a sampling process selecting pairs of different *views* of the same image, obtained by transformation such as cropping, rescaling, or color jittering [Misra and Maaten, 2020, Chen et al., 2020a, He et al., 2020, Grill et al., 2020]. These methods learn to eliminate the irrelevant part of position and color information, in order to satisfy the invariance criterion, and perform well on image classification benchmarks. Some recent approaches go beyond learning global features: to tackle tasks such as semantic segmentation where spatial information plays a key role, [Yang et al., 2021, Xie et al., 2021, Hénaff et al., 2021, Yang et al., 2022, Hénaff et al., 2022, El-Nouby et al., 2022] also learn image models with more emphasis on *local* image structure. In the end most recent approaches to self-supervised learning of visual features learn the corresponding image model using either a (possibly quite sophisticated) global criterion, or a (necessarily) different one exploiting local image characteristics and spatial information. The best performing local methods require a non-parametric pre-processing step that compute unsupervised

36th Conference on Neural Information Processing Systems (NeurIPS 2022).

segmentation masks [Hénaff et al., 2021], which can be done online [Hénaff et al., 2022], but with an additional computational burden.

We argue that more complex reasoning systems should be structured in a hierarchical way, by learning at several scales. To this end, we propose VICRegL, a method that learn features at a global scale, and that additionally uses spatial information, and matches feature vectors that are either pooled from close-by regions in the original input image, or close in the embedding space, therefore learning features at a local scale. In practice, the global VICReg criterion [Bardes et al., 2022] is applied to pairs of feature vectors, before and after the final pooling layer of a convolutional network, thus learning local and global features at the same time. When a segmentation mask is available as in [Hénaff et al., 2021], feature vectors that correspond to the same region in the mask can be pooled together, and compared using a contrastive loss function, which allows spatial vectors far away in the original image to be pooled together if they belong to the same object. In our case, segmentation masks are not available, and we therefore face two challenges: (1) there is no a priori information on how to pool vectors from the same object together, thus long-range matching should be learned in a self-supervised manner, and (2) contrasting negatively feature vectors corresponding to far away locations can have a negative effect, as these vectors could have been pooled from locations that represent the same object in the image. In order to address these issues, VICRegL (1) matches feature vectors according to a $l^2$ nearest-neighbor criterion exploiting both the distances between features and image locations, properly weighted, and (2) uses the VICReg criterion between matched feature vectors. We use VICReg for its simplicity and its *non-contrastive* nature, which alleviates the need for negative samples and therefore does not have an explicit negative contrasting effect between feature vectors that could have been potential matches.

We demonstrate the effectiveness of VICRegL by evaluating the learned representations on vision tasks such as image classification on ImageNet and semantic segmentation on various datasets. Our evaluation is (mostly) done in the setting where the backbone learned by VICRegL is frozen, with only a linear classification or segmentation head tuned to the task at hand. We believe that this setting is a much better evaluation metric than the commonly used fine-tuning benchmarks, as the performance can not be attributed to the use of a complex head, or to the availability of the ground truth masks. Our results show that learning local features, in addition to global features, does not hurt the classification performance, but significantly improves segmentation accuracy. On the Pascal VOC linear frozen semantic segmentation task, VICRegL achieves **55.9** mIoU with a ResNet-50 backbone, which is a **+8.1** mIoU improvement over VICReg, and **67.5** mIoU with a ConvNeXt-S backbone, which is a **+6.6** mIoU improvement.

## 2   Related work

**Global features.** Most recent methods for global feature learning are based on a joint embedding architecture that learns representations that are invariant to various views. These methods differ in the way collapsing solutions are avoided. Contrastive methods [Hjelm et al., 2019, Chen et al., 2020a, He et al., 2020, Chen et al., 2020b, Mitrovic et al., 2021, Dwibedi et al., 2021, Chen et al., 2021, Tomasev et al., 2022] uses negative samples to push dissimilar samples apart from each other. Clustering methods [Caron et al., 2018, 2020, 2021] ensure a balanced partition of the samples within a set of clusters. Non-contrastive methods, which are dual to contrastive ones [Garrido et al., 2022], rely on maintaining the informational content of the representations by either explicit regularization [Zbontar et al., 2021, Bardes et al., 2022] or architectural design [Chen and He, 2020, Grill et al., 2020, Richemond et al., 2020, Lee et al., 2021]. Finally, the best performing methods today are based on vision transformers [Caron et al., 2021, Chen et al., 2021, Li et al., 2022, Zhou et al., 2022a,b] and deliver strong results in both downstream classification and segmentation tasks.

**Local features.** In opposition to global methods, local one focus on explicitly learning a set of local features that describe small parts of the image, which global methods do implicitly [Chen et al., 2022], and are therefore better suited for segmentation tasks. Indeed these methods commonly only evaluate on segmentation benchmarks. A contrastive loss function can be applied directly: (1) at the pixel level [Xie et al., 2021], which forces consistency between pixels at similar locations; (2) at the feature map level [Wang et al., 2021], which forces consistency between groups of pixels: (3) at the image region level [Xiao et al., 2021], which forces consistency between large regions that overlap in different views of an image. Similar to [Wang et al., 2021], our method VICRegL operates at the feature map level but with a more advanced matching criterion that takes into account the distance in

pixel space between the objects. Copy pasting a patch on a random background [Yang et al., 2021, Wang et al., 2022] has also shown to be effective for learning to localize an object without relying on spurious correlations with the background. Aggregating multiple images corresponding to several object instances into a single image can also help the localization task [Yang et al., 2022]. These approaches rely on carefully and handcrafted constructions of new images with modified background or with aggregation of semantic content from several other images, which is not satisfactory, while our method simply rely on the classical augmentations commonly used in self-supervised learning. The best current approaches consist in using the information from unsupervised segmentation masks, which can be computed as a pre-processing step [Hénaff et al., 2021] or computed online [Hénaff et al., 2022]. The feature vectors coming from the same region in the mask are pooled together and the resulting vectors are contrasted between each other with a contrastive loss function. These approaches explicitly construct semantic segmentation masks using k-means for every input image, which is computationally not efficient, and is a strong inductive bias in the architecture. Our method does not rely on these masks and therefore learns less specialized features.

## 3   Method

**Background.** VICReg was introduced as a self-supervised method for learning image representations that avoid the collapse problem by design. Its loss function is composed of three terms: a *variance* term, that preserves the variance of the embeddings, and consists in a hinge loss function on the standard deviation, on each component of the vectors individually and along the batch dimension; an *invariance* term, which is simply an $l^2$ distance between the embeddings from the two branches of a siamese architecture; and finally a *covariance* term, that decorrelates the different dimensions of the embeddings, by bringing to 0 the off-diagonal coefficients of the empirical covariance matrix of the embeddings.

For completeness, we describe how the VICReg framework works [Bardes et al., 2022]. A seed image $I$ is first sampled in the unlabelled training dataset. Two views $x$ and $x'$ are obtained by a rectangular crop at random locations in $I$, rescaling them to a fixed size $(R, S)$ and applying various color jitters with random parameters. The views are fed to an encoder $f_\theta : \mathbb{R}^{C \times R \times S} \to \mathbb{R}^C$ producing their *representations* $y = f_\theta(x)$ and $y' = f_\theta(x') \in \mathbb{R}^C$, which are mapped by an expander $h_\phi : \mathbb{R}^C \to \mathbb{R}^D$ onto the *embeddings* $z = h_\phi(y)$ and $z' = h_\phi(y') \in \mathbb{R}^D$. The VICReg loss function is defined as follows:

$$\ell(z, z') = \lambda s(z, z') + \mu[v(z) + v(z')] + \nu[c(z) + c(z')], \tag{1}$$

where $s$, $v$ and $c$ are the invariance, variance and covariance terms as described in [Bardes et al., 2022], and $\lambda$, $\mu$ and $\nu$ are scalar coefficients weighting the terms.

### 3.1   VICRegL: feature vectors matching

When the encoder $f_\theta$ is a convolutional neural network, the final representations are obtained by performing an average pooling operation $\oplus : \mathbb{R}^{C \times H \times W} \to \mathbb{R}^C$ on the output feature maps, with $C$ the number of channels and $(H, W)$ the spatial dimensions. We now denote the pooled representations $y_\oplus$ and $y'_\oplus \in \mathbb{R}^C$ and the unpooled representations $y$ and $y' \in \mathbb{R}^{C \times H \times W}$. We denote $y_{i,j}$ and $y'_{i,j} \in \mathbb{R}^C$ the feature vectors at position $(i, j)$ in their corresponding feature maps. The main idea is to apply the VICReg criterion between pairs of feature vectors from $y$ and $y'$, by matching an element of $y$ to one from $y'$, using spatial and $l^2$-distance based information. We introduce a local projector network $h_\phi^l : \mathbb{R}^{C \times H \times W} \to \mathbb{R}^{D \times H \times W}$, that embed the feature maps $y$ and $y' \in \mathbb{R}^{C \times H \times W}$ onto feature maps embeddings $z = h_\phi^l(y)$ and $z' = h_\phi^l(y') \in \mathbb{R}^{D \times H \times W}$. The standard expander of VICReg is now the global expander $h_\psi^g : \mathbb{R}^C \to \mathbb{R}^D$, which maps the pooled representations $y_\oplus$ and $y'_\oplus \in \mathbb{R}^C$ to the embeddings $z_\oplus = h_\psi^g(z_\oplus)$ and $z'_\oplus = h_\psi^g(z'_\oplus) \in \mathbb{R}^D$. We describe now how we perform the matching, and introduce our loss functions.

**Location-based matching.** In order to take into account the transformation that occurs between the views $x$ and $x'$ of an image $I$, and thus matching features from similar locations, we compute the absolute position in $I$ that corresponds to the coordinate of each feature vector in its feature map. Each feature vector $z_p$ at position $p$ in the feature map is matched to its spatial nearest neighbor according the the absolute position in $I$, and among the $H \times W$ resulting pairs, only the top-$\gamma$ pairs

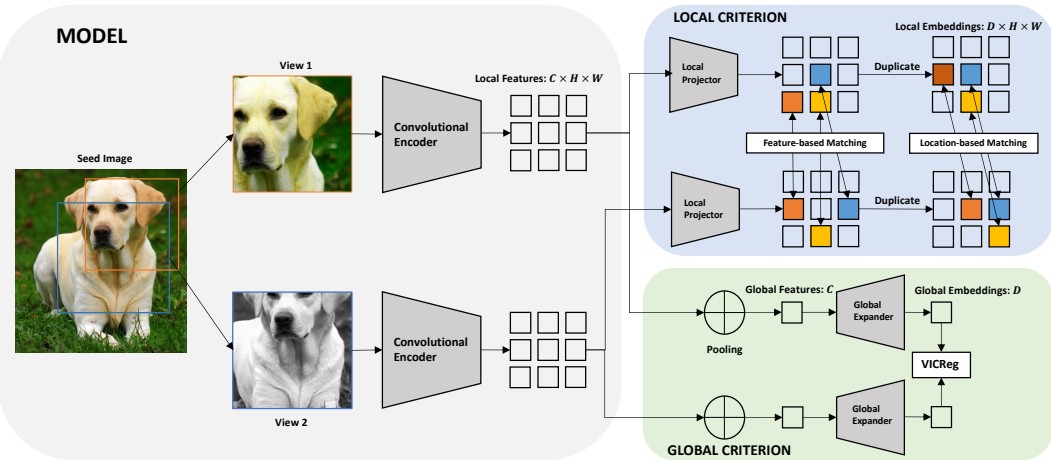

Figure 1: **Overview of VICRegL: Learning local and global features with VICReg.** Given a seed image, two views are produced and fed to an encoder that produces local features. The features are further processed by a local projector that embed them into a smaller space, without destroying the localization information. Two matchings, one based on the spatial information provided by the transformation between the views, the other one based on the $l^2$-distance in the embedding space are computed, and the VICReg criterion is then applied between matched spatial embeddings. Additionally, the local features from the encoder are pooled together, and the pooled features are fed to a global expander. The VICReg criterion is finally applied between the two resulting embeddings.

are kept. The location-based matching loss function is defined as follows:

$$\mathcal{L}_s(z, z') = \sum_{p \in P} l(z_p, z'_{\text{NN}(p)}), \tag{2}$$

where the sum is over coordinates $p$ in $P = \{(h, w) \mid (h, w) \in [1, ..., H] \times [1, ..., W]\}$ the set of all coordinates in the feature map, and $\text{NN}(p)$ denotes the (spatially) closest coordinate $p'$ to $p$ according to the actual distance in the seed image.

**Feature-based matching.** In addition to matching features that are close in terms of location in the original image, we match features that are close in the embedding space. Each feature vector $z_p$ at position $p$ is matched to its nearest neighbor in $z'$ according to the $l^2$ distance in the embedding space, and among the $H \times W$ resulting pairs, only the top-$\gamma$ pairs are kept. The feature-based matching loss function is defined as follows:

$$\mathcal{L}_d(z, z') = \sum_{p \in P} l(z_p, \text{NN}'(z_p)), \tag{3}$$

where the sum is over coordinates $p$ in $P$ and $\text{NN}'(z_p)$ denotes the closest feature vector to $z_p$ in the feature maps $z'$, in terms of the $l^2$-distance. Similar to the location-based loss function, the feature-based loss function enforces invariance on a local scale, but between vectors that are close in the embedding space, and not necessarily pooled from the same location in the seed image. The purpose of this loss function is mainly to capture long-range interactions not captured by the location-based matching.

The general idea of top-$\gamma$ filtering is to eliminate the mismatched pairs of feature vectors that are too far away in the image for the location-based matching, and that therefore probably do not represent the same objects, but most importantly that are probably mismatched for the feature-based matching, especially at the beginning of the training when the network matches feature vectors representing different objects or textures. Sometime, two views don't or barely overlap, for the feature-based matching this is not an issue, as the purpose of this matching is to capture long-range interactions not captured by location-based matching. For the location-based matching, given the parameters we use to generate the views (each view covers between 8% and 100% of the image, chosen uniformly), the probability for the views to not overlap is small, and even in that case matching the closest points between the views does not degrade the final performance. Indeed, we have tried to use a variable

number of matches and a threshold value used to compute the matches, which did not improve the performance compared to using a fixed number of matches. In practice, there are with high probability always good local features to match as the views have a low probability of not overlapping, and this explains why always matching the top-$\gamma$ pairs, compared to introducing a threshold value at which the pair is considered a match, does not degrade the performance.

Our final loss function is a combination of the location-based and feature-based loss functions, which form the local criterion, with in addition a standard VICReg loss function applied on the pooled representations, which is the global criterion. Both location and feature-based loss functions are symmetrized, because for both, the search for the best match is not a symmetric operation. Our final loss function is described as follows:

$$\mathcal{L}(z, z') = \alpha \ell(z_\oplus, z'_\oplus) + (1 - \alpha)\{\mathcal{L}_s(z, z') + \mathcal{L}_s(z', z) + \mathcal{L}_d(z, z') + \mathcal{L}_d(z', z)\}, \quad (4)$$

where $\alpha$ is an hyper-parameter controlling the importance one wants to put on learning global rather than local features. We study later in Section 4.2 the influence of $\alpha$ on the downstream performance, and show that there exists a trade-off such that the learned representations contain local and global information at the same time, and therefore transfer well on both image classification and segmentation tasks.

## 3.2 VICRegL with the ConvNeXt backbone

The feature matching procedure is designed to work with any kind of convolutional neural network. In the experimental section, we provide results with a standard ResNet-50 backbone. However, one can considerably improve the performance on downstream tasks by using a more sophisticated backbone. We propose to use the recently introduced ConvNeXt architecture [Liu et al., 2022], that is very similar to the ResNet one, but with many simple modifications to the original architecture, which make it work as well as modern vision transformers. To the best of our knowledge, this is the first time ConvNeXts have been used in self-supervised learning, and our work shows that convolutional neural networks are still able to deliver state-of-the-art performances in most standard self-supervised learning benchmarks. In order to make the performance competitive with recent approaches, we use the multi-crop strategy introduced in [Caron et al., 2020]. Surprisingly, we found that the combination of multi-crop with encoders from the ResNet family and the VICReg criterion is extremely difficult to optimize, and haven't been able to make it work properly. However, multi-crop shows very good results when combined with ConvNeXts. Our intuitive explanation is based on a study of the optimal batch size. The VICReg criterion regularizes the empirical covariance matrix of the embeddings, computed using the current batch, and we hypothesize that there is a link between the size of the batch, and the dimensionality of the representations. VICReg combined with a ResNet-50 has shown to have an optimal batch size of 2048, which is exactly the dimensionality of the representations of a ResNet-50. We found that the optimal batch size when working with ConvNeXts is 512 which is much smaller, and correlates with the fact that ConvNeXts also have smaller representations (768 for ConvNeXt-S and 1024 for ConvNeXt-B). The optimal batch size might therefore be close to the dimensionality of the representations. Now, the multi-crop strategy artificially increases the size of the batch, which is much easier to handle when the size of the batch before multi-crop is small. Indeed the effective batch size otherwise becomes too large, which causes optimization issues.

We now describe how the matching loss functions are adapted in order to work with multi-crop. Instead of generating only two views of the seed image, $N$ views (2 large, and $N - 2$ small), resized to two different resolutions are generated, and further encoded into the feature maps embeddings $z^1$ and $z^2$ for large views, and $\{z^n\}_{n=3}^N$ for small views. The spatial matching loss function is then defined as follows:

$$\mathcal{L}_s(\{z^n\}_{n=1}^N) = \sum_{m=1}^2 \sum_{n \neq m}^N \{\sum_{p \in P} l(z_p^m, z_{\text{NN}(p)}^n) + \sum_{p \in P} l(z_p^n, z_{\text{NN}(p)}^m)\}, \quad (5)$$

where only the top-$\gamma_1$ and top-$\gamma_2$ pairs of feature vectors of large and small views respectively are kept in the computation of the loss. The feature-based loss function, and the global criterion are adapted in a very similar way, one large view is matched to the other large views and to the small views, and the final loss function is defined as follows:

$$\mathcal{L}(\{z^n\}_{n=1}^N) = \alpha \ell(\{z_\oplus^n\}_{n=1}^N) + (1 - \alpha)\{\mathcal{L}_s(\{z^n\}_{n=1}^N) + \mathcal{L}_d(\{z^n\}_{n=1}^N)\}. \quad (6)$$

Table 1: **Comparison of various *global* and *local* self-supervised learning methods on different linear evaluation benchmarks.** Evaluation of the features learned from a ResNet-50 backbone trained with different methods on: (1) linear classification accuracy (%) (frozen) on the validation set of ImageNet (2) Linear segmentation (mIoU) (frozen and fine-tuning) on Pascal VOC, (3) Linear segmentation (mIoU) (frozen) on Cityscapes. $\alpha$ is the weight of Eq. (4) balancing the importance given to the global criterion, compared to the local criterion. The best result for each benchmark is **bold font**. VICRegL consistently improves the linear segmentation mIoU over the VICReg baseline, which shows that introducing a local criterion is beneficial for a localized understanding of the image.

| | | Linear Cls. (%) | Linear Seg. (mIoU) | | |
| | | ImageNet | Pascal VOC | | Cityscapes |
| Method | Epochs | Frozen | Frozen | Fine-Tuned | Frozen |
|---|---|---|---|---|---|
| *Global features* | | | | | |
| MoCo v2 [Chen et al., 2020b] | 200 | 67.5 | 35.6 | 64.8 | 14.3 |
| SimCLR [Chen et al., 2020a] | 400 | 68.2 | 45.9 | 65.4 | 17.9 |
| BYOL [Grill et al., 2020] | 300 | **72.3** | 47.1 | 65.7 | 22.6 |
| VICReg [Bardes et al., 2022] | 300 | 71.5 | 47.8 | 65.5 | 23.5 |
| *Local features* | | | | | |
| PixPro [Xie et al., 2021] | 400 | 60.6 | 52.8 | 67.5 | 22.6 |
| DenseCL [Wang et al., 2021] | 200 | 65.0 | 45.3 | 66.8 | 11.2 |
| DetCon [Hénaff et al., 2021] | 1000 | 66.3 | 53.6 | 67.4 | 16.2 |
| InsLoc [Yang et al., 2022] | 400 | 45.0 | 24.1 | 64.4 | 7.0 |
| $CP^2$ [Wang et al., 2022] | 820 | 53.1 | 21.7 | 65.2 | 8.4 |
| ReSim [Xiao et al., 2021] | 400 | 59.5 | 51.9 | 67.3 | 12.3 |
| *Ours* | | | | | |
| VICRegL $\alpha = 0.9$ | 300 | 71.2 | 54.0 | 66.6 | 25.1 |
| VICRegL $\alpha = 0.75$ | 300 | 70.4 | **55.9** | **67.6** | **25.2** |

## 3.3 Implementation details

We provide here the implementation details necessary to reproduce the results obtained with our best ResNet-50 and ConvNeXts models. All the models are pretrained on the 1000-class unlabelled ImageNet dataset. Most hyper-parameters are kept unchanged compared to the implementation provided by [Bardes et al., 2022], the VICReg loss variance, invariance and covariance coefficients are set to 25, 25 and 1. The global expander is a 3-layers fully-connected network with dimensions (2048-8192-8192-8192). The local projector is much smaller, due to memory limitations, and has dimensions (2048-512-512-512). With the ResNet-50 backbone, we train our models on 32 Nvidia Tesla V100-32Gb GPUs, with the LARS optimizer [You et al., 2017, Goyal et al., 2017], a weight decay of $10^{-6}$, a batch size of 2048 and a learning rate of 0.1. The learning rate follows a cosine decay schedule [Loshchilov and Hutter, 2017], starting from 0 with 10 warmup epochs and with final value of 0.002. The number of selected best matches $\gamma$ of Eq. (2) and (3) is set to 20. With ConvNeXts backbones, we noticed that much smaller batch sizes actually improve the performance, we therefore train our ConvNeXt-S models on 8 Nvidia Tesla V100-32Gb GPUs, with the AdamW optimizer [Loshchilov and Hutter, 2019], a weight decay of $10^{-6}$, a batch size of 384 and a learning rate of 0.001, and our ConvNeXt-B models on 16 Nvidia Tesla V100-32Gb GPUs with a batch size of 572 and the same other hyper-parameters. The learning rate follows a cosine decay schedule, starting from 0 with 10 warmup epochs and with final value of 0.00001. The number of selected best matches $\gamma_1$ and $\gamma_2$ of Eq. (5) are set to 20 for feature maps from large views and 4 for feature maps from small views.

## 4 Experimental Results

In this section, we evaluate the representations obtained after pretraining VICRegL with a ResNet-50, and ConvNeXt backbones [Liu et al., 2022] of various size, on linear classification on ImageNet-1k [Deng et al., 2009], and linear semantic segmentation on Pacal VOC [Everingham et al., 2010], Cityscapes [Cordts et al., 2016] and ADE20k [Zhou et al., 2019]. We demonstrate that VICRegL strongly improves on segmentation results over VICReg while preserving the classification perfor-

Table 2: **Comparison of various *global* and *local* self-supervised learning methods on different linear evaluation benchmarks.** Evaluation of the features learned from ConvNeXt and ViT backbones trained with different methods on: (1) linear classification accuracy (%) (frozen) on the validation set of ImageNet (2) Linear segmentation (mIoU) (frozen and fine-tuning) on Pascal VOC, (3) Linear segmentation (mIoU) (frozen) on ADE20k. $\alpha$ is the weight of Eq. (4) balancing the importance given to the global criterion, compared to the local criterion. The best result for each benchmark is **bold font**. † denotes pretraining on ImageNet-22k.

| Method | Backbone | Params | Epochs | Linear Cls. (%) ImageNet Frozen | Linear Seg. (mIoU) Pascal VOC Frozen | FT | ADE20k Frozen |
|---|---|---|---|---|---|---|---|
| *Global features* | | | | | | | |
| MoCo v3 [Chen et al., 2021] | ViT-S | 21M | 300 | 73.2 | 57.1 | 75.9 | 23.7 |
| DINO [Caron et al., 2021] | ViT-S | 21M | 400 | 77.0 | 65.2 | 79.5 | 30.5 |
| IBOT [Zhou et al., 2022a] | ViT-S | 21M | 400 | 77.9 | 68.2 | 79.9 | 33.2 |
| VICReg [Bardes et al., 2022] | CNX-S | 50M | 400 | 76.2 | 60.1 | 77.8 | 28.6 |
| MoCo v3 | ViT-B | 85M | 300 | 76.7 | 64.8 | 78.9 | 28.7 |
| DINO | ViT-B | 85M | 400 | 78.2 | 70.1 | 82.0 | 34.5 |
| IBOT [Zhou et al., 2022a] | ViT-B | 85M | 400 | **79.5** | 73.0 | 82.4 | 38.3 |
| MAE [He et al., 2022] | ViT-B | 85M | 400 | 68.0 | 59.6 | 82.4 | 27.0 |
| VICReg | CNX-B | 85M | 400 | 77.6 | 67.2 | 81.1 | 32.7 |
| *Local features* | | | | | | | |
| CP$^2$ [Wang et al., 2022] | ViT-S | 21M | 320 | 62.8 | 63.5 | 79.6 | 25.3 |
| *Ours* | | | | | | | |
| VICRegL $\alpha = 0.9$ | CNX-S | 50M | 400 | 75.9 | 66.7 | 80.0 | 30.8 |
| VICRegL $\alpha = 0.75$ | CNX-S | 50M | 400 | 74.6 | 67.5 | 80.6 | 31.2 |
| VICRegL $\alpha = 0.9$ | CNX-B | 85M | 400 | 77.1 | 69.3 | 81.2 | 33.5 |
| VICRegL $\alpha = 0.75$ | CNX-B | 85M | 400 | 76.3 | 70.4 | 82.5 | 35.3 |
| VICRegL $\alpha = 0.75^†$ | CNX-XL | 350M | 150 | 79.4 | **78.7** | **84.1** | **43.2** |

mance, and is competitive with other local and global self-supervised learning methods. We choose the linear evaluation with frozen weights as our main evaluation metrics, as we believe it is a much better way of evaluating the learned representations. Indeed, the performance can not be attributed to the use of a complex segmentation head, or to the availability of the ground truth masks, and contrary to the frozen setting, the fine-tuning setting measures whether the relevant information is present in the representation, but does not measure if the information is easily extractable from it. We perform the linear evaluation using the protocol introduced by [Zhou et al., 2022a], where the learned feature maps are fed to a linear classifier that outputs a vector with the same size as the number of target classes in the dataset, and is then upsampled to the resolution of the image to produce the predicted mask. The results are averaged over 3 runs with randomly initialized parameters and we found that the difference in performance between worse and best runs is always lower than 0.2%.

## 4.1 Comparison with prior work

**ResNet-50 backbone.** Table 1 presents our results against several other global and local self-supervised learning methods, all pretrained with a ResNet-50 backbone [He et al., 2016]. The main observation we make is the improvement of VICRegL over VICReg on linear segmentation. On Pascal VOC, when the weights of the backbone are frozen, VICRegL $\alpha = 0.9$ improves by **+6.2** mIoU while only loosing 0.3% classification accuracy, and VICRegL $\alpha = 0.75$ improves by **+8.1** mIoU. On fine-tuning the improvement is less significative, which we attribute to the non-informative nature of fine-tuning benchmarks. Indeed, some methods like InsLoc and CP$^2$ that seem competitive on fine-tuning significantly underperform in the frozen regime, which shows that the actual performance of these methods can be attributed to the fact that the weights of the backbone benefit form the availability of the labels during the fine-tuning phase. On Cityscapes, which is much harder, most methods do not perform very well in the linear frozen regime, which sets a new challenge for self-supervised learning of local features. VICRegL outperforms the VICReg baseline by **+1.7** mIoU, as well as every other local features methods by a significant margin. The second observation we make is the robustness of VICRegL in classification, which indicates that it learns both local and

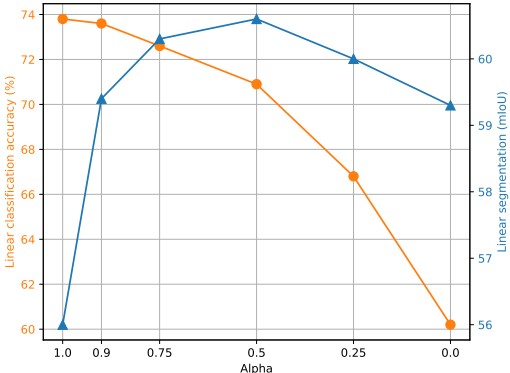

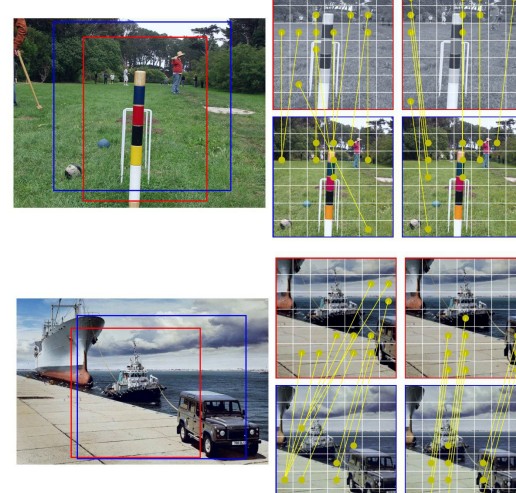

Figure 2: **Study of the trade-off between local and global criteria.** Evaluation on linear classification on ImageNet and on linear Segmentation on Pascal VOC of VICRegL pretrained with various $\alpha$ coefficients of Eq. (4), controlling the importance of the global criterion against the local criterion.

Table 3: **Ablation: matching criterion.** Comparison between using the feature-based matching loss ($\mathcal{L}_d$), the location-based matching loss ($\mathcal{L}_s$), none of the two (Baseline), or both at the same time.

| Method | Cls. (%) | Seg. (mIoU) |
|---|---|---|
| Baseline | **73.8** | 56.0 |
| $\mathcal{L}_s$ | 73.5 | 58.9 |
| $\mathcal{L}_d$ | 73.6 | 57.7 |
| $\mathcal{L}_s$-$\mathcal{L}_d$ | 73.6 | **60.3** |

Figure 3: **Selected matches:** visualization of the locations of the best local matches selected by VICRegL. Left image is the seed image, with in red and blue the crop locations for the two views. Left column are the feature-based matches. Right column are the location-based matches. Only 10 matches are visualized for better clarity, but the actual number of selected matches is 20. We display the matches according to the location of the feature vectors in the feature maps. Note that the receptive field of these feature vectors is much larger than only the patch represented by one square of the grid in the figure. Best viewed in color with **zoom**.

global features at the same time. The performance of most local methods is greatly impacted on classification where they all perform around 10 to 20% below global methods. Global methods on the contrary are efficient for classification but underperform in segmentation compared to VICRegL.

**ConvNeXt backbone.** Table 2 presents our results when pretraining with ConvNeXts backbones against several other global and local self-supervised learning methods pretrained with vision transformers [Dosovitskiy et al., 2021]. Similar to our experiments with a ResNet-50 backbone, the main observation we make is the improvement on segmentation tasks provided by the introduction of the local criterion. With a ConvNeXt-S backbone, in the linear frozen regime, VICRegL $\alpha = 0.9$ improves over VICReg by **+6.6** mIoU on the Pascal VOC, and by **+2.2** mIoU on the ADE20K, while preserving most of the classification accuracy. VICRegL $\alpha = 0.75$ further improves by **+7.4** mIoU and **+3.6** over VICReg on these two benchmarks respectively. With a ConvNeXt-B backbone, the performance improvement remain consistent over VICReg, and VICRegL $\alpha = 0.75$ is competitive with other strong methods such as DINO and IBOT. The improvement also remain consistent in linear fine-tuning where VICRegL also achieves a strong performance. Finally, we report the performance of a much larger ConvNeXt-XL backbone, pretrained in ImageNet-22k, which is significantly improved on segmentation tasks and set a new state-of-the art in linear segmentation. Our results highlight the trade-off between classification and segmentation performance, which can be controlled by the weight given to the local criterion.

## 4.2 Ablations

For all the experiments done in this section, unless specified otherwise, we pretrain a ConvNeXt-S on ImageNet over 100 epochs, with the hyper-parameters described in Section 3.3, and report both the linear classification accuracy on ImageNet, and the linear frozen segmentation mIoU on Pascal VOC.

Table 4: **Ablation: SSL criterion.** Introducing our local criterion with VICReg gives a stronger improvement compared to SimCLR, which is contrastive. (ResNet-50, 300 epochs)

| Method | Cls. (%) | Seg. (mIoU) |
|---|---|---|
| VICReg | **71.1** | 47.8 |
| VICRegL | 70.4 | **55.9** |
| SimCLR | 67.5 | 45.9 |
| SimCLR-L | 66.6 | 51.3 |

Table 5: **Ablation: impact of multi-crop.** Introducing our local criterion yields an improved performance with or without the usage of multi-crop.

| Method | Multi-crop | Cls. (%) | Seg. (mIoU) |
|---|---|---|---|
| VICReg | | 70.1 | 52.9 |
| VICRegL | | 69.9 | 57.8 |
| VICReg | ✓ | **73.9** | 54.4 |
| VICRegL | ✓ | 73.6 | **60.3** |

Table 6: **Ablation: number of selected matches.** The large feature maps are of size $7 \times 7$ and the small ones are of size $3 \times 3$. There are therefore a total number of $49$ large and $9$ small feature vectors and as many possible matches, and only the top-$\gamma_1$ large and top-$\gamma_2$ small are kept.

| $\gamma_1$ | $\gamma_2$ | Cls. (%) | Seg. (mIoU) |
|---|---|---|---|
| 10 | 2 | 73.4 | 59.2 |
| 20 | 4 | **73.6** | **60.3** |
| 49 | 9 | 73.5 | 59.6 |

Table 7: **Ablation: VICReg local criterion.** The collapse problem is automatically prevented with the global criterion. We study here how regularizing the local feature vectors influence the performance. V: variance criterion is used, I: invariance criterion is used, C: covariance criterion is used.

| Criterion | Cls. (%) | Seg. (mIoU) |
|---|---|---|
| I | 73.4 | 59.0 |
| VI | 73.3 | 58.2 |
| VIC | **73.6** | **60.3** |

**Trade-off between the local and global criterion.** The parameter $\alpha$ of Eq. (4) controls the importance that is given to the global criterion, compared to the local criterion. Figure 2 shows that there exists a fundamental trade-off between the ability of a model to learn global visual features, as opposed to learning local features. In the case $\alpha = 1.0$, which is simply VICReg, the model is very efficient at producing global representations of the image, as demonstrated by the performance of 73.9% in classification accuracy. When $\alpha < 1$, which introduces the local criterion, the performance in segmentation is greatly increased, by **+3.4** mIoU when $\alpha = 0.9$, **+4.3** mIoU when $\alpha = 0.75$ and **+4.6** mIoU when the local and global criteria are weighted equally. At the same time, the classification accuracy only drops by respectively 0.2%, 1.2% and 2.9%. This highlights the existence of a sweet spot, where the model is strongly performing at both classification and segmentation, which indicates that it has learned both meaningful local and global features. When $\alpha$ decreases too much, the model starts to lose its performance in both tasks, which shows that having a global understanding of the image is necessary, even for localized tasks.

**Study of the importance between feature-based and location-based local criteria.** VICRegL matches feature vectors according to a location-based criterion $\mathcal{L}_s$ of Eq. (2) and a feature-based criterion $\mathcal{L}_d$ of Eq. (3). Table 3 study the importance of these criterion. Baseline in the table means that no local criterion is used, and is simply VICReg. The location-based criterion gives the best improvement by **+2.9** mIoU over the baseline, compared to only **+1.7** mIoU for the feature-based criterion, but it is the combination of the two that significantly improves over the baseline by **+4.3** mIoU, which shows that using both the learned distance in the embedding space in combination with the actual distance in the pixel space produces local features with the best quality. In all cases, the classification accuracy is not affected, which is expected as the local criterion has little effect on the quality of the global features, and therefore on the downstream classification accuracy.

**Study of the number of matches.** We study here the influence of changing the number of selected best matches $\gamma_1$ and $\gamma_2$ of Eq. (5), to keep for the computation of the local losses. For our experiments with multi-crop, the size of the feature maps is $(2048 \times 7 \times 7)$ for the large crops and $(2048 \times 3 \times 3)$ for the small crops. There are therefore in one branch of the siamese architecture 49 feature maps for large crops, and 9 for small crops. Tables 6 shows that there is a trade-off between keeping all the matches ($\gamma_1 = 49$ and $\gamma_2 = 9$), and keeping a small number of matches ($\gamma_1 = 10$ and $\gamma_2 = 2$), and that the best segmentation performance is obtained with an in-between number of matches, ($\gamma_1 = 20$ and $\gamma_2 = 4$), which improves by **+0.7** mIoU compared to keeping all the matches. Similar to the study on the influence of the local losses, the classification accuracy is not affected, as the local criterion does not improve or degrade the global features.

**Study of VICReg components for the local criterion.** The global criterion is sufficient for the vectors to not collapse to trivial solutions. We therefore investigate if introducing the variance (V) and

covariance (C) criterion, in addition to the invariance (I) criterion, applied by the local loss functions on the feature vectors, is useful or not. Table 7 shows that these regularization criteria are actually helping the performance, introducing the variance criterion improves on segmentation by **+0.8** mIoU, and additionally adding the covariance criterion further improves the performance by **+1.3** mIoU over the baseline. Similar to other ablations on the local loss functions, the classification accuracy is not significantly impacted.

**Study of a different collapse prevention method.** Our collapse-prevention mechanism is the variance and covariance regularization of VICReg, which is a non-contrastive criterion that therefore does not contrast negatively on potential positive matchings. We study however the incorporation of our local criterion with a contrastive criterion, SimCLR [Chen et al., 2020a]. We simply replace VICReg by SimCLR in Eq. (2) and Eq. (3) and refer this new method as SimCLR-L. Table 4 reports the performance, and we observe that although there is a gap in performance between regular SimCLR and VICReg, the additional benefit provided by the local criterion is much stronger with VICReg.

**Impact of multi-crop.** We study the impact of using the multi-crop strategy for the data augmentation, by comparing VICReg to VICRegL. Table 5 reports our results. Whether multi-crop is used or not, introducing the local criterion always improve significantly the segmentation results, while preserving again most of the classification performance.

### 4.3 Visualization

We provide in Figure 3, a visualization of the pairs of matched feature vectors selected by VICRegL. Right to the seed image, the left column shows the feature-based matches, and the right column shows the location-based matches. Each case in the the grid represents a position in the feature map, and a match between two feature vectors is represented by a yellow line. The receptive field of these feature vectors is larger than a single case in the grid, and actually spans the entire image, but we observe that the embedding space is shaped such that the feature-based matching is coherent regarding the semantic content at a position in the image where a feature vector is pooled. A feature vector that is located at a position corresponding to a texture representing "sky" or "grass" in one view is matched to another one on the other view located at a position corresponding to a similar "sky" or "grass" texture. Additional visualizations are available in Appendix **??**.

## 5  Conclusion

In this work, we introduced VICRegL, a method for learning both local and global visual features at the same time, by matching feature vectors with respect to their distance in the pixel space and in the embedding space. We show that introducing a local criterion significantly improves the performance on segmentation tasks, while preserving the classification accuracy. We also demonstrate that convolutional networks are competitive to vision transformers in self-supervised learning, by using the ConvNeXt backbone.

**Limitations and Future work.** Convolutional neural networks by design produce feature maps that have a receptive field that covers the entire image. It is not clear to which extent a feature vector at a given position in the feature maps actually contains mainly information about the objects located at the corresponding location in the input image. The learned tokens of a vision transformers are also good candidates for local features, and a detailed study of the actual local nature of both the feature vectors of a convolutional network and the tokens of a vision transformer, would provide useful insights for future directions of self-supervised learning of local features. Future work will also tackle the problem of learning hierarchical features, by applying a criterion not only at a local and a global scale, but also at multiple levels in the encoder network.

**Acknowledgement.** Jean Ponce was supported in part by the French government under management of Agence Nationale de la Recherche as part of the "Investissements d'avenir" program, reference ANR-19-P3IA-0001 (PRAIRIE 3IA Institute), the Louis Vuitton/ENS Chair in Artificial Intelligence and the Inria/NYU collaboration. Adrien Bardes was supported in part by a FAIR/Prairie CIFRE PhD Fellowship. The authors wish to thank Li Jing, Randall Balestriero, Amir Bar, Grégoire Mialon, Jiachen Zhu, Quentin Garrido, Florian Bordes, Bobak Kiani, Surya Ganguli, Megi Dervichi, Yubei Chen, Mido Assran, Nicolas Ballas and Pascal Vincent for useful discussions.

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
