**Algorithm 1:** VICRegL pytorch pseudocode.

```
1 # f: encoder network, lambda, mu, nu: coefficients of the invariance,
      variance and covariance losses, N: batch size, D: dimension of the
      representations, H: height of the feature maps, W: width of the feature
      maps, alpha: coefficient weighting the global and local criteria
2 # mse_loss: Mean square error loss function, off_diagonal: off-diagonal
      elements of a matrix, relu: ReLU activation function, l2_dist: l2-distance
      between two vectors, avg_pool: average pooling, filter_best: filter to
      only keep the gamma best matches
3
4 for x in loader: # load a batch with N samples
5     # two randomly augmented versions of x
6     x_a, x_b, pos_a, pos_b = augment(x)
7
8     # compute representations
9     map_a = f(x_a) # N x C x H x W
10    map_b = f(x_b) # N x C x H x W
11
12    # Loss computation
13    local_loss = local_criterion(map_a, map_b, pos_a, pos_b)
14    global_loss = vicreg(avg_pool(map_a), avg_pool(map_b))
15    loss = alpha * global_loss + (1 - alpha) * local_loss
16
17    # optimization step
18    loss.backward()
19    optimizer.step()
20
21 def local_criterion(map_a, map_b, pos_a, pos_b):
22    # Location-based loss function
23    map_a_nn = torch.zeros_like(map_a)
24    for h, w in H, W:
25        h',w' = argmin(l2_dist(pos_a[h,w], pos_b[h',w']))
26        map_a_nn[h, w] = map_b[h',w']
27
28    map_a_filtered, map_a_nn_filtered = filter_best(map_a, map_a_nn, gamma)
29    location_loss = vicreg(map_a_filtered, maps_a_nn_filtered)
30
31    # Feature-based loss function
32    map_a_nn = torch.zeros_like(map_a)
33    for h, w in H, W:
34        h',w' = argmin(l2_dist(map_a[h,w], map_b[h',w']))
35        map_a_nn[h, w] = maps_b[h',w']
36
37    map_a_filtered, map_a_nn_filtered = filter_best(map_a, map_a_nn, gamma)
38    feature_loss = vicreg(map_a_filtered, map_a_nn_filtered)
39
40    return location_loss + feature_loss
41
42 def vicreg(z_a, z_b):
43    # invariance loss
44    sim_loss = mse_loss(z_a, z_b)
45
46    # variance loss
47    std_z_a = torch.sqrt(z_a.var(dim=0) + 1e-04)
48    std_z_b = torch.sqrt(z_b.var(dim=0) + 1e-04)
49    std_loss = torch.mean(relu(1 - std_z_a)) + torch.mean(relu(1 - std_z_b))
50
51    # covariance loss
52    z_a = z_a - z_a.mean(dim=0)
53    z_b = z_b - z_b.mean(dim=0)
54    cov_z_a = (z_a.T @ z_a) / (N - 1)
55    cov_z_b = (z_b.T @ z_b) / (N - 1)
56    cov_loss = off_diagonal(cov_z_a).pow_(2).sum() / D
57                + off_diagonal(cov_z_b).pow_(2).sum() / D
58
59    return lambda * sim_loss + mu * std_loss + nu * cov_loss
```

Table 8: **Comparison of various *global* and *local* self-supervised learning methods on frozen and finetune evaluation with a non-linear FCN head.** Evaluation of the features learned from a ResNet-50 backbone pretrained with different methods on frozen and fine-tuning segmentation of a FCN head [Long et al., 2015] on Pascal VOC. $\alpha$ is the weight of Eq. (4) balancing the importance given to the global criterion, compared to the local criterion. The best result for each benchmark is **bold font**.

| | | FCN Seg. (mIoU) | |
| Method | Epochs | Frozen | Fine-Tuned |
|---|---|---|---|
| *Global features* | | | |
| MoCo v2 [Chen et al., 2020b] | 200 | 60.5 | 66.1 |
| SimCLR [Chen et al., 2020a] | 400 | 62.7 | 69.5 |
| BYOL [Grill et al., 2020] | 300 | 62.0 | 69.7 |
| VICReg [Bardes et al., 2022] | 300 | 58.9 | 65.8 |
| *Local features* | | | |
| PixPro [Xie et al., 2021] | 400 | 61.7 | 70.1 |
| DenseCL [Wang et al., 2021] | 200 | 63.8 | 69.4 |
| DetCon [Hénaff et al., 2021] | 1000 | **66.1** | 70.0 |
| InsLoc [Yang et al., 2022] | 400 | 58.1 | 69.2 |
| CP$^2$ [Wang et al., 2022] | 820 | 57.9 | 66.9 |
| ReSim [Xiao et al., 2021] | 400 | 62.7 | 70.3 |
| *Ours* | | | |
| VICRegL $\alpha = 0.9$ | 300 | 62.3 | 68.5 |
| VICRegL $\alpha = 0.75$ | 300 | 65.3 | **70.4** |

## A  Additional results

**Evaluation with a non-linear head.** We report in Table 8 a comparison between various methods with a non-linear FCN head [Long et al., 2015] both in the frozen and the fine-tuning regime. VICRegL significantly outperforms the other methods expect DetCon in the frozen regime, and all the methods in the fine-tuning regime. We observe that the performance gap between method is much closer with a non-linear head, compared to the lienar head, which favors again the use of the linear head for downstream segmentation evaluation of self-supervised features.

**Evaluation on object detection.** Table 9 reports our results on detection downstream tasks. We follow DenseCL [Wang et al., 2021] and fine-tune a Faster R-CNN detector [Ren et al., 2015] (C4-backbone) on the Pascal VOC trainval07+12 set with standard 2x schedule and test on the VOC test2007 set (Fine-tune in the Table). Additionally, we provide results using our linear frozen benchmark, that we adapt to produce bounding boxes instead of segmentation heads (Linear frozen in the Table). With a ResNet-50 backbone, VICRegL offers better performance than other methods in the fine-tuning regime, and shines in the frozen regime with an improvement of **+3.7 AP** over VICReg and of **+2.6 AP** over DenseCL. With the ConvNeXt-B backbone, VICRegL outperforms concurrent methods using a ViT-B backbone by a significant margin.

**Semi-supervised evaluation.** Table 10 report our results on the semi-supervised evaluation benchmark proposed by [Wang et al., 2021]. We report APb on detection and APm on segmentation on COCO. VICRegL significantly outperforms the other methods on these benchmarks.

## B  Additional visualizations

We provide in Figure 4 and Figure 5, additional visualizations that show the matches that VICRegL selects for its local criterion. In most examples, we see that the feature-based matching is able to match regions that are from very far away locations but that represents a similar concept. For example, the left ea. or leg of an animal is matched to its right ear; similar textures are matched; and when there is twice the same object in the image, both are matched together. Some matches seem not relevant, this is explained again by the fact that the feature vector attention span is larger than just the square represented in the figures, with convolution networks it is actually covering the entire image.

Table 9: **Comparison of various *global* and *local* self-supervised learning methods on frozen and finetune detection downstream evaluations.** Evaluation of the features learned from a ResNet-50 backbone trained with different methods on: (1) fine-tuning detection of a Faster-RCNN head [Ren et al., 2015] on Pascal VOC; (2) frozen detection with a linear head. The best result for each benchmark is **bold font**.

| Method | Backbone | Fine-Tuned (AP) | Linear Det. (AP) |
|---|---|---|---|
| MoCo v2 [Chen et al., 2020b] | R50 | 57.0 | 41.3 |
| VICReg [Bardes et al., 2022] | R50 | 57.4 | 41.9 |
| DenseCL [Wang et al., 2021] | R50 | 58.7 | 43.0 |
| VICRegL $\alpha = 0.75$ | R50 | 59.5 | 45.6 |
| Moco-v3 [Chen et al., 2020b] | ViT-B | 69.8 | 54.1 |
| VICReg [Bardes et al., 2022] | ViT-B | 71.3 | 55.4 |
| DINO [Wang et al., 2021] | ViT-B | 74.5 | 58.4 |
| VICRegL $\alpha = 0.75$ | CNX-B | **75.4** | **59.0** |

Table 10: **Comparison of various *global* and *local* self-supervised learning methods on frozen and finetune detection downstream evaluations.** Evaluation of the features learned from a ResNet-50 backbone trained with different methods on: (1) fine-tuning detection of a Faster-RCNN head [Ren et al., 2015] on Pascal VOC; (2) frozen detection with a linear head. The best result for each benchmark is **bold font**.

| Method | Detection (APb) | Segmentation (APm) |
|---|---|---|
| MoCo v2 [Chen et al., 2020b] | 23.8 | 20.9 |
| VICReg [Bardes et al., 2022] | 24.0 | 20.9 |
| DenseCL [Wang et al., 2021] | 24.8 | 21.8 |
| VICRegL $\alpha = 0.75$ | **25.7** | **22.6** |

## C  Method details

We give more details on how the corresponding location in the seed image of a feature vector is computed. Given a feature map $y \in \mathbb{R}^{C \times H \times W}$ containing $H \times W$ feature vectors of dimension $C$, we start with a matrix $A$ of size $H \times W$, where the coefficient at position $(i, j)$ in $A$ is the the coordinate of the center of a rectangle at position $(i, j)$ in a $H \times W$ grid of rectangles of identical width and height, such that the size of the grid is similar to the size of the the crop $x$ made in the original image $I$. From $A$ which contains relative coordinates regarding the upper left corner of the crop $x$ corresponding to $y$, we compute the position matrix $P$ of same size which contains the corresponding absolute coordinates regarding the upper left corner of $I$. If $x$ is an horizontally flipped version of $I$, we permute the values in $P$ accordingly. The matrix $P$ can then be used for the nearest neighbors search in the spatial loss function of Eq. (2).

## D  Evaluation details

We use our own scripts for the linear evaluations and the mmsegmentation library [Contributors, 2020] for all our segmentation evaluations. For most methods used in our comparison of Table 1 and 2, the linear classification and segmentation results are not always available. We therefore download available pretrained models from the official repository of each method respectively, and perform the evaluations ourselves by sweeping extensively over hyper-parameters, in particular the learning rate, for each model and each downstream task.

### D.1  Linear Evaluation

We follow common practice and use the same procedure as most recent methods [Caron et al., 2020, Grill et al., 2020, Zbontar et al., 2021, Bardes et al., 2022], and train a linear classifier on the representations obtained with the frozen backbone. We use the SGD optimizer, a batch size of 256, and train for 100 epochs. We report the best result using a learning rate among 30, 10, 1.0,

Table 11: **Running time and peak memory.** Comparison between VICReg and VICRegL for pretraining a ResNet-50 backbone with a batch size of 2048. The training is distributed on 32 Tesla V100 GPUs, the running time is measured over 100 epochs and the peak memory is measured on a single GPU.

| Method | time / 100 epochs | peak memory / GPU |
|--------|-------------------|-------------------|
| VICReg | 11h | 11.3G |
| VICRegL | 13h | 15.7G |

0.3, 0.1, 0.03, 0.01. The augmentation pipeline also follows common practices, at training time the images are randomly cropped and resized to $224 \times 224$. At validation time the images are resized to $256 \times 256$ and center cropped to size $224 \times 224$. No color augmentation are used for either training and evaluation.

## D.2  Linear Segmentation

**ResNet-50 backbone.** We train a linear layer on the representations obtained from the backbone (frozen or fine-tuned), to produce segmentation masks. For all dataset, the images are of size $512 \times 512$. Given the output feature map of dimension $(2048, 32, 32)$ of a ResNet-50, the map is first fed to a linear layer that outputs a map of dimension $(num\_classes, 32, 32)$, which is then upsampled using bilinear interpolation to the predicted mask of dimension $(num\_classes, 512, 512)$. We use the *40k iterations schedule* available in mmsegmentation with both Pascal VOC and Cityscapes, with the SGD optimizer, and we pick the best performing learning rate among 0.1, 0.05, 0.03, 0.02, and 0.01.

**ConvNeXts and ViTs backbones.** We train a linear layer on the representations at various layers from the backbone (frozen or fine-tuned), to produce segmentation masks. For all dataset, the images are of size $512 \times 512$. For ConvNeXts the output feature maps of the four blocks are of dimension $(128, 128, 128)$, $(256, 64, 64)$, $(512, 32, 32)$ and $(1024, 16, 16)$. We upsample all the feature maps to $(x, 128, 128)$ and concatenate them into a tensor of dimension $(1920, 128, 128)$, which is fed to a linear layer that outputs a map of dimension $(num\_classes, 128, 128)$. For ViTs the output feature maps of the four blocks are all of dimension $(768, 32, 32)$. We concatenate them into a tensor of dimension $(3072, 32, 32)$, which is fed to a linear layer that outputs a map of dimension $(num\_classes, 32, 32)$. The final map is upsampled using bilinear interpolation to the predicted mask of dimension $(num\_classes, 512, 512)$. We use the *40k iterations schedule* available in mmsegmentation with Pascal VOC and the *160k iterations schedule* with ADE20k, with the AdamW optimizer, and we pick the best performing learning rate among $1e - 03$, $8e - 04$, $3e - 04$, $1e - 04$, and $8e - 05$ for the frozen regime, and $1e - 04$, $8e - 05$, $3e - 05$, $1e - 05$, $8e - 06$, $3e - 06$, and $1e - 06$ for the fine-tuned regime.

## E  Running time and memory usage

We report in Table 11, the running time of VICRegL in comparison with VICReg, for pretraining a ResNet-50 backbone with a batch size of 2048. The introduction of the local criterion comes with an additional computation overhead, mainly do to computing the covariance matrices of every feature vector in the output feature maps. These computations induces a moderate additional computational burden both in terms of time and memory usage.

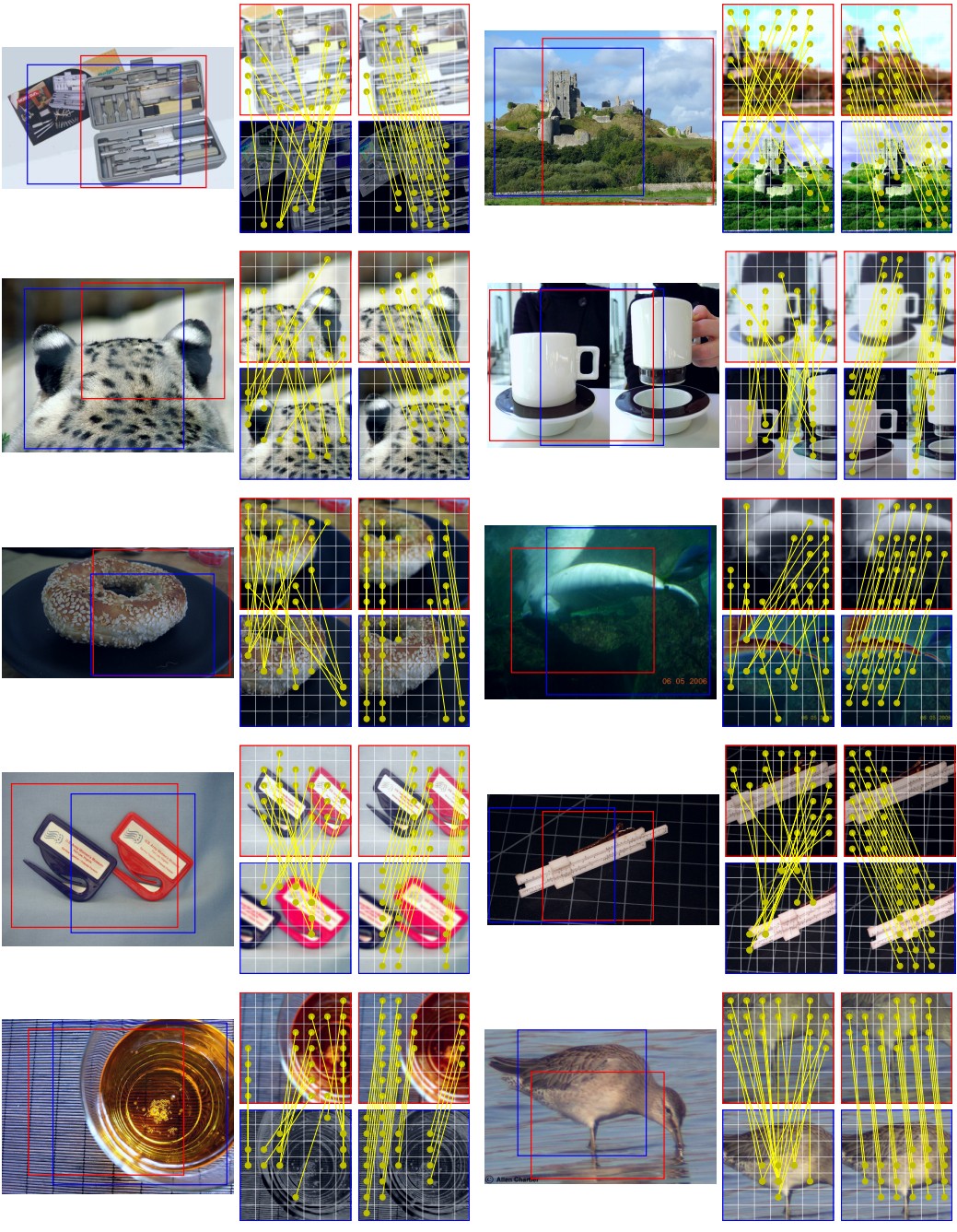

Figure 4: **Selected matches:** visualization of the locations of the best local matches selected by VICRegL. Left image is the seed image, with in red and blue the crop locations for the two views. Left column are the $l^2$-distance based matches. Right column are the location based matches. Only 10 matches are visualized for better clarity, but the actual number of selected matches is 20. We display the matches according to the location of the feature vectors in the feature maps. Note that the receptive field of these feature vectors is much larger than only the patch represented by one square of the grid in the figure. Best viewed in color with **zoom**.

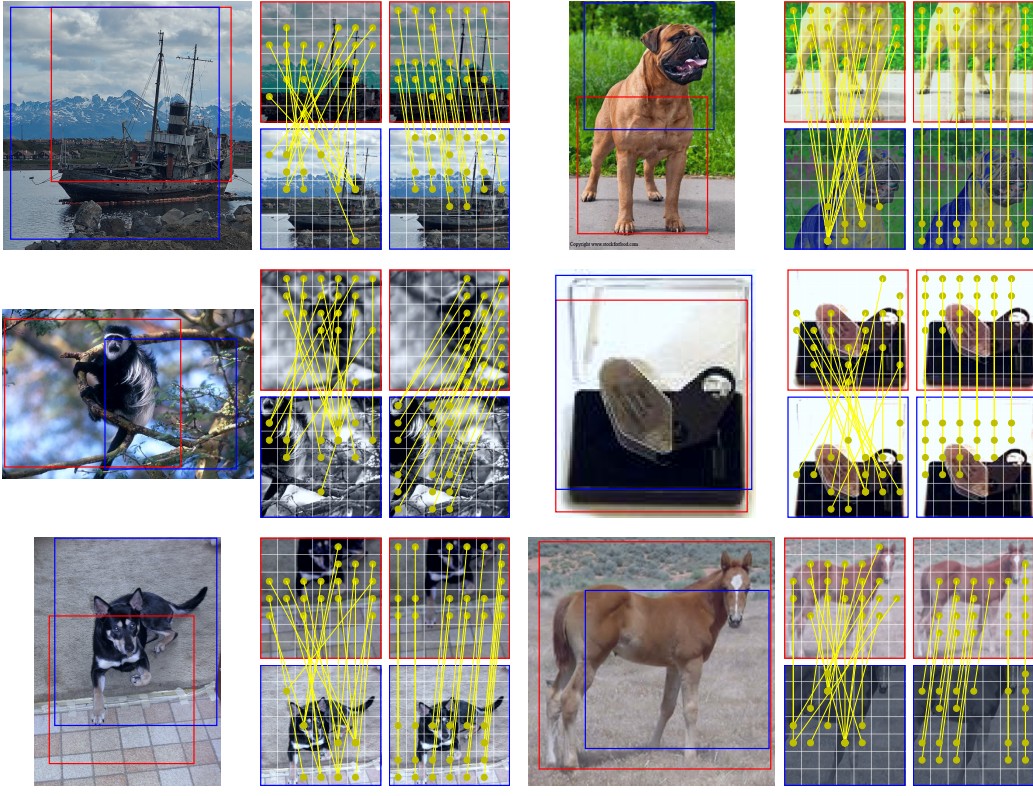

Figure 5: **Selected matches:** visualization of the locations of the best local matches selected by VICRegL. Left image is the seed image, with in red and blue the crop locations for the two views. Left column are the $l^2$-distance based matches. Right column are the location based matches. Only 10 matches are visualized for better clarity, but the actual number of selected matches is 20. We display the matches according to the location of the feature vectors in the feature maps. Note that the receptive field of these feature vectors is much larger than only the patch represented by one square of the grid in the figure. Best viewed in color with **zoom**.