# OpenReview forum: "VICRegL: Self-Supervised Learning of Local Visual Features"
_NeurIPS.cc/2022/Conference — NeurIPS 2022 Accept_

### Official Review · Reviewer_Pb2x · 2022-07-05

**Rating:** 7
**Confidence:** 4
**Soundness:** 3 good
**Presentation:** 4 excellent
**Contribution:** 2 fair

**Summary:**

The paper describes a learning method to learn jointly global and local descriptors in a self-supervised way.

The global descriptor are learned following VICREG [1], a learning method that tackles the issue of mode collapse during representation learning. Two versions of the same images are derived using random transformations (crop, color jittering) and fed to a siamese network. Each branch produces a 1D feature vector representing the image. The branches are optimized separately with the VICREG constraints: the feature vector must be close in the embedding space (invariance loss), the standard deviation of the embedding dimensions must be close to one (variance loss), and the dimensions of the embeddings should not be correlated (covariance loss).

The local descriptor learning is the main contribution of the paper. The feature maps provide local descriptors (a feature map of dimension H x W x C can be seen as a set of HW local features of dimension C). These descriptors are optimized so that: 1) descriptors from the same location in the image are pushed together (spatial matching loss) 2) descriptors close in the embedding space are also pushed close together (even when they do not relate to the same image location i.e. they are not ground truth match).

The several components of the final loss are balanced with metaparameters investigated in the paper. The main finding are: 1) that the bigger weight given to the global loss, the worse local features are for object detection and semantic segmentation tasks; 2) the global criterion is relevant to improve the local features suggesting that they benefit from a global understanding of the image.

The paper evaluates the global descriptors on classification and local descriptors on object detection and semantic segmentation. It compares against relevant instances of the state of the art and experiments show that the introduced optimization is indeed relevant to learn local features in a self-supervised way.

[1] Vicreg: Variance-invariance-covariance regularization for self-supervised learning

**Questions:**

L49: "contrasting feature vectors corresponding to far away locations can have a negative effect, as these vectors could have been pooled from locations that represent the same object in the image."
This is not clear. Does contrasting here mean pushing the feature vector together? In this case, wouldn't one want to push features of the same object together? i.e. this would be a positive outcome?

The l2-distance (Eq 3) is not intuitive at first. It pushes together local features close in the embedding space even though they might not be ground-truth matches. Is there any intuition on why this is beneficial (as supported by the experiments)?
One intuition is that it might act as regularizer or an implicit clustering.
Another is that the global criterion already pushes positive matches coarsely together and this loss refines the vector. Is there any observations or intuition on that interesting phenomenon?

**Limitations:**

- L107: It would be nice to introduce the notation of $\lambda, \mu, \nu$ in Eq 1 for completeness.

- L118: $h_{\phi}^l$ is called a 'local projector' whereas it is the equivalent of the VICREG expander but at the scale of the feature map. Since 120 uses the terminology 'global expander', it would seem more consistent to use the word 'local expander' rather than projector. This would also be more consistent with VICREG on which the paper relies on.

- L127, L135, 181: 'only the top-$\gamma$ pairs are kept'.  It is not clear whether for each feature position, only the top-$\gamma$ are kept to derive the loss, or if the top-$\gamma$ among all pairs in the batch are kept (assuming top-$\gamma$ means the pairs with the smallest distance). This would help the reader if the definition were introduced L127.

- L127, L135, 181: Is there a motivation / intuition on why it is better to keep only the top-$\gamma$ pairs?
Together with the l2-distance loss, this could reinforce the intuition that the l2-distance loss act as a regularizer/booster as it would be applied only to feature vectors that are already close to each other, which one can assume to be ground-truth positives.

- L306: a feature maps -> a feature map

**Strengths And Weaknesses:**

Strengths:
- The problem tackled is relevant to the community: the joint learning of global and local features
- The originality of the l2 distance loss: it pushes feature vector that are close in the embedding space even though they are not positive matches.
- Relevant state-of-the art
- Relevant experiments (comparisons, analysis and ablation)
- Very well-written paper: the organisation and presentation of the ideas is easy to follow, and the description and analysis of the experimental results are clear and valuable to the reader.

Weaknesses:
- The main contribution seems incremental with respect to VICREG.

---

> ### Author Response · Authors · 2022-08-02
> **Official answer to Reviewer Pb2x 1/2**
>
> 1) *The main contribution seems incremental with respect to VICREG.*
>
> Our paper introduces a notion of locality to self-supervised learning (SSL), which can be combined to any SSL criterion other than VICReg, such as SimCLR. The choice of VICReg for the main method is due to its simplicity and its non-contrastive nature, which we believe is more appropriate when matching local descriptors within the same image, as explained in point 2) of our answer to reviewer yFUm. The second main contribution of our paper is to demonstrate empirically the fundamental trade-off there is between learning local and global features, highlighted by Figure 2 of the paper. We will make sure to clarify our contributions in the revision.
>
> 2) *L49: "contrasting feature vectors corresponding to far away locations can have a negative effect, as these vectors could have been pooled from locations that represent the same object in the image." This is not clear. Does contrasting here mean pushing the feature vector together? In this case, wouldn't one want to push features of the same object together? i.e. this would be a positive outcome?*
>
> Here contrasting the feature vectors means pushing them away from each other. We will clarify the sentence.
>
> 3) *The l2-distance (Eq 3) is not intuitive at first. It pushes together local features close in the embedding space even though they might not be ground-truth matches. Is there any intuition on why this is beneficial (as supported by the experiments)? One intuition is that it might act as regularizer or an implicit clustering. Another is that the global criterion already pushes positive matches coarsely together and this loss refines the vector. Is there any observations or intuition on that interesting phenomenon?*
>
> The goal of the feature-based matching (FBM) of Eq. (3) is to capture long-range interactions not captured by the location-based matching (LBM) of Eq.(2). Indeed, LBM will only match vectors that are spatially close in the image, but will ignore potential matches of similar objects far away in the image, FBM is not spatially constrained and can therefore match feature vectors pooled from far away locations in the image. This is illustrated in Figure 3, in the top image, where we can see a feature-based match between two areas both representing grass textures in their respective view, but are far away in the seed image. At the beginning of the training, it is true that FBM matches can be considered as bad matches, because the encoder has not learned good features yet. However, LBM matches are ground-truth matches which allows the system to not fall into degenerated solutions. The global loss also helps not fall into these solutions. At the end of the training, FBM matches are of good quality and capture long-range interactions, as illustrated in Figure 3 of the paper. We will provide additional visualization in the appendix of the revision.
>
> 4) *L107: It would be nice to introduce the notation of \lambda, \mu, \nu in Eq 1 for completeness.*
>
> We will add a sentence mentioning these parameters.
>
> 5) *L118: $h$ is called a 'local projector' whereas it is the equivalent of the VICREG expander but at the scale of the feature map. Since 120 uses the terminology 'global expander', it would seem more consistent to use the word 'local expander' rather than projector. This would also be more consistent with VICREG on which the paper relies on.*
>
> We called it projector as it projects the feature vectors into a space of smaller dimensionality. However we agree that the terminology is confusing, especially because the main reason we do that is because of GPU memory constraint. We will change for the term "local expander".
>
> 6) *L127, L135, 181: 'only the top-\gamma pairs are kept'. It is not clear whether for each feature position, only the top-\gamma are kept to derive the loss, or if the top-\gamma among all pairs in the batch are kept (assuming top-\gamma means the pairs with the smallest distance). This would help the reader if the definition were introduced L127.*
>
> From two views of an example, the corresponding two feature maps are produced by the encoder, and for each feature vector in the first feature map, the best feature vector in the other feature map is computed, which results in N matches if there are N feature vectors. Among these N matches, the top-\gamma best matches are kept. We will clarify the explanation in the text.

---

> > ### Author Response · Authors · 2022-08-02
> > **Official answer to Reviewer Pb2x 2/2**
> >
> > 7) *L127, L135, 181: Is there a motivation / intuition on why it is better to keep only the top- pairs? Together with the l2-distance loss, this could reinforce the intuition that the l2-distance loss act as a regularizer/booster as it would be applied only to feature vectors that are already close to each other, which one can assume to be ground-truth positives.*
> >
> > As explained in point 7) of our answer to reviewer yFUm and point 2) of our answer to reviewer hQQN, the general idea is to eliminate the mismatched pairs of feature vectors:
> > - That are too far away in the image for the location-based matching, and that therefore probably do not represent the same objects.
> > - But most importantly that are probably mismatched for the feature-based matching, especially at the beginning of the training when the network matches feature vectors representing different objects or textures.
> > We will clarify this point in the revision.

---

### Official Review · Reviewer_hQQN · 2022-07-10

**Rating:** 5
**Confidence:** 4
**Ethics Flag:** Yes
**Soundness:** 3 good
**Presentation:** 3 good
**Contribution:** 2 fair

**Summary:**

This paper introduces locality constraints building upon the existing VICReg method. It does so by not destroying the features maps and adding VICReg loss terms on the features maps. Two simple methods of correspondence generation are proposed: a) Nearest neighbour in spatial location, and b) embedding similarity of local features. The paper claims to learn visual representation that improves performances on localized tasks such as detection and segmentation while maintaining the consistently good performance for global classification task. The paper uses two backbones for evaluations and shown to perform better than existing methods especially on linear evaluations. The paper has a good contribution, but a few evaluations and analysis seem lacking.

**Questions:**

1. The method chooses location representation correspondences based on nearest location and embedding similarity. Any thought or analysis on noisy or spurious correspondences? For example, although nearest neighbours the patch may be coming from different object/semantics/texture. Similarly, based on embedding similarity, the network may choose unreliable pairs especially in the early stage of training.

2. In the ablation of selected pairs, when all possible matches are selected, I believe both spatial and embedding distance losses will be the same, and network would try to match all local features though coming from different objects. Though not optimal, it seems the performance is improved compared to the baselines. Please could you provide intuition on this scenario and related improvement.
3.  Is the network trained from scratch with global and local losses or, a pretraining with global loss is done?
4. How does the method compared to [1]

[1] He, Kaiming, et al. "Masked autoencoders are scalable vision learners." Proceedings of the IEEE/CVF Conference on Computer Vision and Pattern Recognition. 2022.

The authors have given satisfactory answers to the rebuttal. I would like to increase my rating to borderline accept given the authors promise to include all discussions and a complete result for detection on PASCAL and MS-COCO with standard evaluation protocol.

**Limitations:**

1.  [Comment/Suggestion] It is natural to fine-tune the network when supervisions are available, however, the gap in the performance boosts narrows after finetuning. May be a semi-supervised task and evaluation would highlight the importance of the method and a more meaningful scenario.
2. It would be interesting to see any sort of analytical study to support the hypothesis of relation of batch size and dimension of the network’s output. This would be an important contribution on itself.

**Strengths And Weaknesses:**

Strength:
1. The idea is elegant, uses the existing loss but additionally on local features.
2. It is shown to significantly improve segmentation performance (though difference in finetuning performances not huge.) which is convincing.
3. Ablations studies show insights to design choices, visualization on matches.
4. Overall, the paper is easy to follow, and the paper has a good contribution however there are a few concerns as follows


Weakness:
1. The main concern is that although the paper claims to benefit detection and segmentation tasks, only segmentation is evaluated.  Evaluations on detection are studied in most of the baseline papers but this is lacking in the paper. It may require a more comprehensive experiment to validate the effectiveness.
2. The authors use ‘multicrop’ technique but do not ablate this factor in the experiments. Also, it is not clear if other baselines (especially VICReg) also enjoy the benefit of multicrop. If they do not then, the comparison may not be fair. The authors may want to clarify this.
3. As the method uses features maps and losses on based on them, it would be interesting to see its impact on training speed and memory.

Please read below on other queries and comments below.

---

> ### Author Response · Authors · 2022-08-02
> **Official answer to Reviewer hQQN 1/2**
>
> 1) *The main concern is that although the paper claims to benefit detection and segmentation tasks, only segmentation is evaluated. Evaluations on detection are studied in most of the baseline papers but this is lacking in the paper. It may require a more comprehensive experiment to validate the effectiveness.*
>
> We agree that detection results are missing and are actually important in order to validate our approach. We therefore provide below experimental results on detection tasks. We follow DenseCL and fine-tune a Faster R-CNN detector (C4-backbone) on the Pascal VOC trainval07+12 set with standard 2x schedule and test on the VOC test2007 set (Fine-tune in the Table). Additionally, we provide results using our linear frozen benchmark, that we adapt to produce bounding boxes instead of segmentation heads (Linear frozen in the Table). We report our results in the following table:
>
> |				| Fine-tune	(AP)				| Linear Frozen (AP)|
> | --- | --- | --- |
> | MoCov2		|	57.0						| 41.3 |
> | VICReg		|	57.4						| 41.9 |
> | DenseCL		|	58.7						| 43.0 |
> | VICRegL 0.75	|   59.5						| 45.6 |
>
> Here again, VICRegL offers better performance than other methods in the fine-tuning regime, but really shines in the frozen regime, with an improvement of +3.7 AP over VICReg and of +2.6 AP over DenseCL. We provide results on the same benchmark with the ConvNeXt-B backbone, in comparison with other competitive methods using the ViT-B backbone.
>
> |				| Fine-tune	(AP)				| Linear Frozen (AP)|
> | --- | --- | --- |
> | MoCo-v3		|	69.8						| 54.1 |
> | VICReg		|	71.3						| 55.4 |
> | DINO			|	74.5						| 58.4 |
> | VICRegL 0.75	|   75.4						| 59.0 |
>
> Again, VICRegL outperforms concurrent methods by a significant margin. These results will be added to our revision.
>
>
> 2) *The authors use ‘multicrop’ technique but do not ablate this factor in the experiments. Also, it is not clear if other baselines (especially VICReg) also enjoy the benefit of multicrop. If they do not then, the comparison may not be fair. The authors may want to clarify this.*
>
> We provide an ablation on the use of multi-crop in point 8) of our answer to reviewer yFUm. We confirm that all methods compared in Table 1 **do not use** multi-crop (including VICReg and our VICRegL), and that all methods in Table 2 **use** multi-crop.
>
> 3) *As the method uses features maps and losses based on them, it would be interesting to see its impact on training speed and memory.*
>
> We provide a comparison between VICReg and VICRegL in terms of running time and memory use, using a ResNet-50 Runs on 32GPUs with a batch size of 2048.
>
> | 	Methods		| RT (100ep Hours)	| MEM (Peak G on GPU) |
> | --- | --- | --- |
> | VICReg		| 11h 	|			11.3G
> | VICRegL		| 13h	|			15.7G
>
> The computation of multiple covariance matrices for each feature vector indeed induces a moderate additional computational burden both in terms of time and memory usage.
>
>
> 4) *The method chooses location representation correspondences based on nearest location and embedding similarity. Any thought or analysis on noisy or spurious correspondences? For example, although nearest neighbors the patch may be coming from different objects/semantics/texture. Similarly, based on embedding similarity, the network may choose unreliable pairs especially in the early stage of training.*
>
> This is a good point, and this is the reason why we introduced the top-k best matches selection, which prevents bad pairs from being matched. Our ablation in Table 4 shows how selecting only the best-k helps the performance. Additionally, as explained in point 7) of our answer to reviewer yFUm, there are with high probability always good local features to match as the views have a low probability of not overlapping, and this is why always matching the top-k pairs, compared to introducing a threshold value at which the pair is considered a match, does not degrade the performance. We will clarify this point in the revision.

---

> > ### Author Response · Authors · 2022-08-02
> > **Official answer to Reviewer hQQN 2/2**
> >
> > 5) *In the ablation of selected pairs, when all possible matches are selected, I believe both spatial and embedding distance losses will be the same, and network would try to match all local features though coming from different objects. Though not optimal, it seems the performance is improved compared to the baselines. Please could you provide intuition on this scenario and related improvement.*
> >
> > Selecting all the matches in Table 4 does not mean that all N^2 pairs are selected, it means that for all N feature vectors in view 1, we search for the NN in view 2 and match the two. Therefore different matching criteria give different matchings, which is the case here, with the location-based criterion, which only captures short-range interactions, and the feature-based criterion, which is able to capture long-range interactions. We will clarify this point in the revision.
> >
> > 6) *Is the network trained from scratch with global and local losses or, a pretraining with global loss is done?*
> >
> > The network is trained from scratch with both the global and local losses, but pretraining with only the global loss and fine-tuning with the local loss is an interesting idea that should be explored by future work.
> >
> > 7) *How does the method compare to [1]*
> >
> > We provide a comparison of our VICRegL model with a ConvNeXt-B backbone against MAE with a ViT-B backbone [1]. The ConvNeXt-B and ViT-B models both have the same number of parameters. We report results on linear and fine-tuning classification on ImageNet, linear and fine-tuning segmentation on Pascal VOC, and fine-tuning segmentation on ADE20k with a UperNet head.
> >
> > |	Method |	ImageNet Classification |  |	PascalVOC Segmentation    | ADE20k Segmentation |
> > | --- | --- | --- | --- | --- |
> > |					| Linear Frozen (%)	| Linear FT (%) | Linear FT (mIoU)   | UperNet FT (mIoU)|
> > | MAE				|	68.0	| 83.6		| 59.9 | 82.5     | 48.1 |
> > | VICRegL 0.75		| 76.3	 | 83.5		| 70.4 | 82.5     | 48.2
> >
> > VICRegL offers similar results in the fine-tuning regime, but strongly improves over MEA in the frozen regime. We will incorporate MEA as a concurrent method in Table 2 of our revision.
> >
> >
> > 8) *It is natural to fine-tune the network when supervisions are available, however, the gap in the performance boosts narrows after finetuning. May be a semi-supervised task and evaluation would highlight the importance of the method and a more meaningful scenario.*
> >
> > We provide below results on the semi-supervised evaluation benchmark proposed in DenseCL (Table 3). We report APb on detection and APm on segmentation on COCO:
> >
> > | Method		| Detection	(APb)					| Segmentation (APm)|
> > | --- | --- | --- |
> > | MoCov2		| 23.8							| 20.9 |
> > | VICReg		| 24.0							| 20.9 |
> > | DenseCL		| 24.8							| 21.8 |
> > | VICRegL 0.75	| 25.7							| 22.6 |
> >
> > VICRegL significantly outperforms the other methods on these benchmarks. These results will be included in the revision.
> >
> > 9) *It would be interesting to see any sort of analytical study to support the hypothesis of relation of batch size and dimension of the network’s output. This would be an important contribution on itself.*
> >
> > We provide in point 8) of our answer to reviewer yFUm a possible explanation for the fact that ResNet-50 combined with multi-crop does not give strong performance, however the relation between batch size and dimensionality of the network's output is still unclear. We have conducted experiments that empirically show a boost in downstream performance when the size of the batch is equal or close to the dimensionality of the output. However a more theoretical study is required.

---

### Official Review · Reviewer_yFUm · 2022-07-11

**Rating:** 4
**Confidence:** 4
**Soundness:** 3 good
**Presentation:** 2 fair
**Contribution:** 2 fair

**Summary:**

The paper proposes VICRegL, a method that extends the existing (global) VICReg objective with local contrastive objectives where the loss is applied at the level of feature pixels (between spatial locations in the feature map before global pooling in CNNs).
Two approaches to building pairs of feature pixels for contrastive learning are proposed: 1) based on the l2 distance in the learned embedding space, and 2) based on the known spatial location between the two generated views.
In experiments, the method is evaluated on (global) image classification tasks on ImageNet and (local) semantic segmentation tasks on Pascal VOC, Citiscapes, and ADEK20K.
Ablations demonstrate a trade-off between global and local performance. A good balance between global and local SSL objectives can obtain competitive image classification performance while achieving SotA performance on segmentation tasks.

**Questions:**

I would appreciate it if the authors could address the weakness listed above, especially the first four points, i.e.,
- How does the method compare to [N1]?
- Is the VICReg necessary, or would the approach also work with other objectives?
- How are the results for prior work obtained in Table 1, and what explains the inconsistency with DenseCL?
- How does the work relate to and differ from the various related prior works (including [N1-N3])?

**Limitations:**

Limitations were addressed well in the paper. The point about whether feature pixels learn local information is interesting and could be explored through experiments, e.g., by clustering the feature pixels of an image and visualizing these clusters.


**Strengths And Weaknesses:**

Strengths:
- Study of SSL pre-training that performs well on various downstream tasks (local and global) is valuable
- Good performance in transfer to both classification and segmentation
- Technical presentation is mostly clear
- A good set of ablation experiments is provided, demonstrating:
	- the trade-off between local and global feature learning
	- The two ways to construct local pairs (l2 in embedding space and spatial distance)
	- The influence of the number of local pairs to use in the loss


Weaknesses:
- The approach to building positive local pairs via spatial distance is very similar to [N1], which is neither discussed in the paper nor compared to in experiments.
- The method is demonstrated only with VICReg as the contrastive criterion. It is unclear if the approach to incorporate local objectives with the global one would also work with other "contrastive" objectives. In principle, it should also work, and demonstrating it with more standard contrastive objectives (e.g., SimCLR) would strengthen the paper.
- Inconsistencies with results reported in prior work: DenseCL by Wang et al. reports 69.4 mIoU on PascalVOC with fine-tuning. It is unclear how the difference from the reported 66.8 in Table 1 comes about.
- Several related works on learning both global and local representations are summarized in Section 2, but it is unclear how the proposed method relates to and differs from these works.
- The idea of using nearest-neighbors in feature space for contrastive learning has been explored in prior works (e.g., [N2, N3]). While it is used in a different context here, these works should be discussed.
- The paper argues for learning "hierarchical features" (L39), however, it is unclear how the proposed method should learn more hierarchical features than existing approaches based on CNN architectures.
- The number of local pairs is fixed in the method. What happens, however, if the two views don't or barely overlap?
- The paper mentions that multi-crop was used with ConvNeXt architectures. Therefore, providing an ablation with and without multi-crop in this setting would be good. Also, it would be good to investigate more the reason for multi-crop failing with ResNet-50, e.g., test smaller batch sizes as hypothesized in Section 3.2.
- Minor: Error in L104 - It should not be C in both input and output


[N1] Xiao et al.: "Region Similarity Representation Learning", ICCV2021

[N2] Dwibedi et al.: "With a little help from my friends: Nearest-neighbor contrastive learning of visual representations."

[N3] Koohpayegani et al.: "Mean Shift for Self-Supervised Learning"

---

> ### Author Response · Authors · 2022-08-02
> **Official answer to Reviewer yFUm 1/3**
>
> 1) *The approach to building positive local pairs via spatial distance is very similar to [N1], which is neither discussed in the paper nor compared to in experiments.*
>
> We were not aware of this paper and the proposed ReSIM method. We will make sure to mention it both in the text and the experimental results of our paper. Our method VICRegL, differs from ReSIM on a number of points:
>
> * Contrary to ReSIM which focuses on location-based matching that only captures short range interactions, VICRegL combines both a location-based matching and an feature-based matching, which allows for capturing both short-range and long-range interactions within an image.
>
> * ReSIM matches the overlapping region of two views of a seed image by averaging the feature vectors within some sliding window. VICRegL directly compares the feature vectors themselves and can match areas coming from non-overlapping regions of the views.
>
> * ReSIM uses a contrastive criterion. VICRegL uses a non-contrastive criterion that empirically performs better, and does not push away similar but unmatched local features. This is a key component in handling long-range interactions where similar objects or textures are matched from far away locations in the image.
>
> * ReSIM focuses on learning local image features and is evaluated on vision tasks that best exploit them, such as object detection and instance segmentation. We argue instead that they should be combined with more global features useful in tasks such as image categorization. We show that there is a fundamental tradeoff between the two types of  features and focus on optimizing this tradeoff.
>
> Finally, we incorporate an experimental comparison with ReSIM in point 3) of our answer to reviewer yFUm, and show that VICRegL significantly outperforms ReSIM on several downstream tasks.
>
>
>
> 2) *The method is demonstrated only with VICReg as the contrastive criterion. It is unclear if the approach to incorporate local objectives with the global one would also work with other "contrastive" objectives. In principle, it should also work, and demonstrating it with more standard contrastive objectives (e.g., SimCLR) would strengthen the paper.*
>
> We use VICReg for its *non-contrastive* nature, which avoids pulling away from each other features from different regions in the image, but associated with the same object. We observed in practice that VICReg gives a stronger improvement on segmentation downstream tasks when combined with our local criterion. But our method can be used with other criteria as well and we provide here a table where we compare the performance on linear segmentation on Pascal VOC for VICReg, VICRegL, SimCLR, and SimCLR combined with our local criterion (SimCLR-L), pretrained with a ResNet-100 backbone on 100 epochs:
>
> | Method | Linear Segmentation (mIoU) |
> | ----   | ---- |
> |VICReg 	| 47.8 |
> |VICRegL 	| 55.9 |
> |SimCLR 	| 45.9 |
> |SimCLR-L 	| 51.3 |
>
> VICReg performs already better than SimCLR (+1.9 mIoU), and the improvement of VICRegL over VICReg is of +8.1 mIoU compared to only +5.4 mIoU for SimCLR-L over SimCLR. VICRegL also performs better than SimCLR-L with an improvement of +4.6 mIoU. We will include these results in the revision.
>
>
> 3) *Inconsistencies with results reported in prior work: DenseCL by Wang et al. reports 69.4 mIoU on PascalVOC with fine-tuning. It is unclear how the difference from the reported 66.8 in Table 1 comes about.*
>
> The performance of 69.4 reported in [1] is the fine-tuning performance of an **FCN head** (CNN head with multiple layers). Table 1 of our paper reports the fine-tuning performance of a **linear head**, which explains the difference. We report below a comparison including DenseCL and ReSIM with both a linear and a FCN head in the frozen and the fine-tuning regime:
>
> | Method |		Linear Frozen (mIoU)	| FCN Frozen (mIoU)	| Linear FT (mIoU)     | FCN FT (mIoU) |
> | --- | --- | --- | --- | --- |
> | MoCov2			| 35.6	|		41.3	|	64.8  |       67.5 |
> | DenseCL			| 45.3	|		50.4	|	66.8   |      69.4 |
> | ReSIM			| 51.9		|	52.9		| 67.1     |     69.7 |
> | VICRegL $\alpha$=0.75	| 55.9		|	56.0		| 67.6     |    70.3 |
>
> VICRegL significantly outperforms the other methods across all the benchmarks. We will clarify this point and incorporate the FCN results in our revision.

---

> > ### Author Response · Authors · 2022-08-02
> > **Official answer to Reviewer yFUm 2/3**
> >
> > 4) *Several related works on learning both global and local representations are summarized in Section 2, but it is unclear how the proposed method relates to and differs from these works.*
> >
> > Regarding global methods, VICRegL introduces an additional notion of locality and builds on the VICReg method that only learns at a global level. VICRegL therefore falls into the category of non-contrastive methods that maintain the informational content of the learnt features. Regarding local methods, we propose to rewrite the related work section as follows:
> >
> > """
> > Local features. In opposition to global methods, local one focus on learning a set of local features that describe small parts of the image, and are therefore better suited for segmentation tasks. Indeed these methods commonly only evaluate on segmentation benchmarks. A contrastive loss function can be applied directly: (1) at the pixel level [Xie et al., 2021], which forces consistency between pixels at similar locations; (2) at the feature map level [Wang et al., 2021], which forces consistency between groups of pixels: (3) at the image region level [Xiao et al., 2021], which forces consistency between large regions that overlap in different views of an image. Similar to [Wang et al., 2021], our method VICReg operates at the feature map level but with a more advanced matching criterion that takes into account the distance in pixel space between the objects. Copy pasting a patch on a random background [Yang et al., 2021, Wang et al.,2022] has also shown to be effective for learning to localize an object without relying on spurious correlations with the background. Aggregating multiple images corresponding to several object instances into a single image can also help the localization task [Yang et al., 2022]. These approaches rely on carefully and handcrafted constructions of new images with modified background or with aggregation of semantic content from several other images, which is not satisfactory, while our method simply rely on the classical augmentations commonly used in self-supervised learning. The best current approaches consist in using the information from unsupervised segmentation masks, which can be computed as a pre-processing step [Hénaff et al., 2021] or computed online [Hénaff et al., 2022]. The feature vectors coming from the same region in the mask are pooled together and the resulting vectors are contrasted between each other with a contrastive loss function. These approaches explicitly construct semantic segmentation masks using k-means for every input image, which is computationally not efficient, and is a strong inductive bias in the architecture. Our method does not rely on these masks and therefore learns less specialized features.
> > """
> >
> >
> > 5) *The idea of using nearest-neighbors in feature space for contrastive learning has been explored in prior works (e.g., [N2, N3]). While it is used in a different context here, these works should be discussed.*
> >
> > We agree that these approaches should be discussed and we will mention them in the paper. However, the nearest neighbors of [N2] and [N3] are very different from ours. Indeed, these methods search for the closest example from a target global representation of a full image. In our case, one image is decomposed into a set of local descriptors, and we search for the nearest descriptor within this set, using location and feature-based matching.
> >
> >
> >
> > 6) *The paper argues for learning "hierarchical features" (L39), however, it is unclear how the proposed method should learn more hierarchical features than existing approaches based on CNN architectures.*
> >
> > We agree that the word "hierarchical" is not really appropriate here. What we meant is that learning local and global representations can be seen as learning at two different scales. One can imagine learning at more than two scales, by enforcing a local criterion at every layer in an encoder. This future architecture is really what we meant by "hierarchical". We will clarify this in the revision.

---

> > > ### Author Response · Authors · 2022-08-02
> > > **Official answer to Reviewer yFUm 3/3**
> > >
> > > 7) *The number of local pairs is fixed in the method. What happens, however, if the two views don't or barely overlap?*
> > >
> > > For the feature-based matching this is not an issue, as the purpose of this matching is to capture long-range interactions not captured by location-based matching. For the location-based matching, given the parameters we use to generate the views (each view covers between 8% and 100% of the image, chosen uniformly), the probability for the views to not overlap is small, and even in that case matching the closest points between the views does not degrade the final performance. Indeed, we also have tried to use a variable number of matches and a threshold value used to compute the matches, which did not improve the performance compared to using a fixed number of matches. We will discuss this point in the revision.
> > >
> > > 8) *The paper mentions that multi-crop was used with ConvNeXt architectures. Therefore, providing an ablation with and without multi-crop in this setting would be good. Also, it would be good to investigate more the reason for multi-crop failing with ResNet-50, e.g., test smaller batch sizes as hypothesized in Section 3.2.*
> > >
> > > We provide here a comparison between using multi-crop or not with the ConvNeXt architecture. We pre-train a ConvNeXt-S on 100 epochs on ImageNet with VICReg and VICRegL and evaluate on linear frozen segmentation on Pascal VOC.
> > >
> > > | 	Method |			M-C	(mIoU)	| no M-C (mIoU) |
> > > | --- | --- | --- |
> > > | VICReg 			 | 60.1	| 58.7 |
> > > | VICRegL $\alpha$=0.75	 | 67.5	| 64.8 |
> > >
> > > The improvement between VICReg and VICRegL is significant whether multi-crop is used or not. In Table 2 of our paper, all competing methods use multi-crop.
> > >
> > > Regarding the poor performance of the ResNet-50 backbone combined with multi-crop (MC), we have conducted additional experiments, and found that ResNet-50 with MC was strongly overfitting. The ConvNeXt architecture uses stochastic depth residual connection (SDRC) for preventing overfitting, which helps a lot for handling MC. In the MC regime, we found that incorporating SDRC in ResNet-50 helps a lot, especially when using small batch size, which experimentally verifies our hypothesis of Section 3.2, and that ConvNeXt without SDRC will result in strong overfitting.
> > >
> > >
> > >
> > > 9) *Minor: Error in L104 - It should not be C in both input and output*
> > >
> > > We will fix this typo by replacing "C" by "3" in L105.
> > >
> > > 10) *Questions*
> > >
> > > * *How does the method compare to [N1]?*
> > >
> > > See the Table and explanation provided in point 3) of our answer to reviewer yFUm.
> > >
> > > * *Is the VICReg necessary, or would the approach also work with other objectives?*
> > >
> > > See our comparison provided in point 2) of our answer to reviewer yFUm.
> > >
> > >
> > > * *How are the results for prior work obtained in Table 1, and what explains the inconsistency with DenseCL?*
> > >
> > > See our explanation in point 3) of our answer to reviewer yFUm.
> > >
> > > * *How does the work relate to and differ from the various related prior works (including [N1-N3])?*
> > >
> > > See our explanation in points 3), 4) and 5) of our answer to reviewer yFUm

---

### Author Response · Authors · 2022-08-02
**General comments**

We thank the reviewers for their comments about the paper. We have incorporated additional experimental results in this rebuttal, mainly results on new benchmarks (object detection, semi-supervised segmentation, fine-tuning with large heads) for evaluation and comparison with various other methods (VICReg, ReSIM, DenseCL, MAE), as well as an additional ablation of multi-crop, and a study on the running time and memory usage of our method. These new results will be incorporated in the paper. To avoid ambiguities, we use in this rebuttal "feature-based matching" instead of the term "l2-distance based matching" used in the original submission to clearly distinguish this form of matching based on distance between features in some high-dimensional embedding space from "location-based matching" based on the image distance between pixels. The metric used in both cases is the l2 (Euclidean) distance. We will use this terminology in the revision as well.

---

### Meta-Review · Area_Chair_iiUE · 2022-08-29

**Recommendation:** Accept
**Confidence:** Less certain

**Metareview:**

This paper proposes to extend the existing VICReg objective to the local features for obtaining good performances on both image-level and dense prediction tasks. In specific, while the global features are obtained by an average pooling on the output feature maps, the local pairs are determined by both of the feature distance and spatial location distance. The technical novelty seems to be somewhat incremental due to a little bit simple modification of the existing global objective to the local objective for dense representation learning.  However, extensive experiments on several benchmarks including ablations and visualization clearly demonstrate the effectiveness of the proposed self-supervised representation learning for both classification and segmentation (+detection) tasks. Especially, the authors faithfully addressed most concerns and questions raised by the reviewers, and the overall quality of the paper seems to be significantly improved. Therefore, I would recommend to accept this paper.

**Award:**

No

---

### Decision · Program_Chairs · 2022-09-14

Accept